# Establishment of a Screening Method for Epstein-Barr Virus-Associated Gastric Carcinoma by Droplet Digital PCR

**DOI:** 10.3390/microorganisms7120628

**Published:** 2019-11-29

**Authors:** Takuya Shuto, Jun Nishikawa, Kanami Shimokuri, Ayaka Yanagi, Tatsuya Takagi, Fumiya Takagi, Osamu Miura, Michihisa Iida, Hiroaki Nagano, Yoshihiro Takemoto, Eijiro Harada, Yutaka Suehiro, Takahiro Yamasaki, Takeshi Okamoto, Isao Sakaida

**Affiliations:** 1Faculty of Laboratory Science, Yamaguchi University Graduate School of Medicine, 1-1-1 Minami-Kogushi, Ube, Yamaguchi 755-8505, Japan; i002up@yamaguchi-u.ac.jp (T.S.); kshimo@yamaguchi-u.ac.jp (K.S.); i012up@yamaguchi-u.ac.jp (A.Y.); i004up@yamaguchi-u.ac.jp (T.T.); f-takagi@yamaguchi-u.ac.jp (F.T.); 2Hofu Institute of Gastroenterology, 14-33 Ekiminami-machi, Hofu, Yamaguchi 747-0801, Japan; miura@hofu-icho.or.jp; 3Department of Gastroenterological, Breast and Endocrine Surgery, Yamaguchi University Graduate School of Medicine, 1-1-1 Minami-Kogushi, Ube, Yamaguchi 755-8505, Japan; miida@yamaguchi-u.ac.jp (M.I.); hnagano@yamaguchi-u.ac.jp (H.N.); 4Department of Clinical Science of Surgery, Yamaguchi University Graduate School of Medicine, 1-1-1 Minami-Kogushi, Ube, Yamaguchi 755-8505, Japan; ytake@yamaguchi-u.ac.jp (Y.T.); eharada@yamaguchi-u.ac.jp (E.H.); 5Department of Oncology and Laboratory Medicine, Yamaguchi University Graduate School of Medicine, 1-1-1 Minami-Kogushi, Ube, Yamaguchi 755-8505, Japan; ysuehiro@yamaguchi-u.ac.jp (Y.S.); t.yama@yamaguchi-u.ac.jp (T.Y.); 6Department of Gastroenterology and Hepatology, Yamaguchi University Graduate School of Medicine, 1-1-1 Minami-Kogushi, Ube, Yamaguchi 755-8505, Japan; tokamoto@yamaguchi-u.ac.jp (T.O.); sakaida@yamaguchi-u.ac.jp (I.S.)

**Keywords:** Epstein-Barr virus, gastric carcinoma, droplet digital PCR, endoscopic biopsy

## Abstract

Background: Epstein-Barr virus-associated gastric carcinoma (EBVaGC) is classified as one of the molecular subtypes of gastric cancer. We used droplet digital polymerase chain reaction (ddPCR) to enable highly sensitive and quantitative detection of EBV. Methods: EBV-DNA load was calculated based on the copy number of the BamH1-W fragment of EBV by ddPCR, and the cut-off value of EBV-DNA load was set. We conducted both ddPCR and EBER1 ISH to examine whether their results coincided in 158 gastric cancer specimens of unknown EBV status. We prepared 26 biopsy specimens and 49 serum samples including EBVaGC and assayed them by ddPCR. Results: The median values of EBV-DNA load for EBVaGC and EBV-negative control were 17.0 and 0.00308, respectively. A cut-off value of 0.032 was determined for which the sensitivity was 1. Among the 158 gastric cancer specimens, 14 lesions were judged as EBV-positive by the 0.032 cut-off value determined by ddPCR. The results of ddPCR and EBER1 ISH were in complete agreement. Even when using a biopsy specimen as a sample for ddPCR, the EBV-DNA load of all EBVaGCs was larger than the cut-off value. Conclusions: We established a new method of diagnosing EBVaGC from tissue samples by ddPCR.

## 1. Introduction

The Epstein-Barr virus (EBV) is a double-stranded DNA virus. It belongs to the herpes virus family, which latently infects B lymphocytes in adults. EBV is closely associated with both lymphoid and epithelial malignancies, such as Burkitt lymphoma, Hodgkin disease, immunocompromised lymphoma, and nasopharyngeal carcinoma [1,2]. The EBV genome was first detected in gastric cancer by using a polymerase chain reaction (PCR) in 1990 [3]. Since then, gastric cancer patients identified as EBV-positive have been reported to be about 10% of gastric cancer patients in the world [4].

The Cancer Genome Atlas (TCGA) of gastric adenocarcinomas developed a novel classification system dividing gastric cancer into four molecular subtypes: (1) EBV-associated gastric cancer (EBVaGC); (2) microsatellite instability-high (MSI-H); (3) chromosomal instability; and (4) genomically stable tumors. The characteristics of EBVaGC are reported to include the harboring of recurrent PIK3CA (phosphatidylinositol 3-kinase) mutations, extreme DNA hypermethylation, and PD-L1 (programmed cell death ligand 1) and PD-L2 overexpression [5]. Therefore, demethylating agents, PI3K inhibitors, and immune checkpoint inhibitors may be effective in the treatment of EBVaGC [6]. Selection of therapy based on the mechanism of development of EBVaGC is required, and it is important to diagnose EBVaGC prior to treatment.

An in situ hybridization (ISH) method for EBV-encoded small RNA1 (EBER1) has commonly been used to detect EBV. EBVaGC is defined as a gastric carcinoma showing EBER1 signals in the nuclei of almost all carcinoma cells detected by ISH [7,8], however, EBER-ISH is expensive and time consuming and has not been applied to preoperative diagnosis. Real-time quantitative PCR was applied to examine the association between the copy number of EBV-DNA and the clinical courses of EBV-associated diseases [9,10,11]. Thus, we used droplet digital PCR (ddPCR) [12] as a novel method to enable the highly sensitive and quantitative detection of EBV.

## 2. Materials and Methods 

### 2.1. Clinical Materials

We have a pool of cases diagnosed as EBVaGC by EBER1 ISH. EBER1 signals were observed in the nucleus of almost all cancer cells in these EBVaGCs. The cut-off value of EBV-DNA load was set based on 47 lesions of EBVaGC and 47 lesions of EBV-negative gastric cancer, which were matched as much as possible by age, sex, histologic type, and depth of tumor invasion. To evaluate the validity of the cut-off value, we conducted both ddPCR and EBER1 ISH to examine whether their results coincided in 158 specimens of gastric cancer of unknown EBV status (Table 1). These series of gastric cancer cases were treated in the Hofu Institute of Gastroenterology from 2007 to 2010. We prepared 26 biopsy specimens: 21 specimens obtained from EBVaGC and 5 specimens obtained from EBV-negative controls (Appendix A). We also assayed 49 serum samples by ddPCR of which 25 samples were taken from EBVaGC patients preoperatively and 24 samples were taken from patients with EBV-negative gastric cancer (Appendix A).

This study was approved by the Institutional Review Board of Yamaguchi University Hospital (approval number: H30-125-1).

### 2.2. EBV-DNA Load by ddPCR

DNA was isolated using the QIAamp DNA FFPE Tissue kit (QIAGEN, Hilden, Germany). For serum samples, we used 0.4 mL of each sample for DNA extraction with the MagNA Pure Compact Nucleic Acid Isolation Kit I (Roche, Tokyo, Japan) according to the manufacturer’s instructions. We eluted DNA in a volume of 50 μL of elution buffer and quantified it by Qubit 2.0 fluorometers (Thermo Fisher Scientific, Yokohama, Japan). Because the size of the tissue section is different for each sample, the EBV-DNA load was calculated by dividing the copy number of the BamH1-W fragment of EBV by the copy number of telomerase reverse transcriptase (TERT) for normalization between specimens. We performed ddPCR to count the absolute copy numbers of EBV-DNA and TERT [13]. The PCR reaction was performed with 40 ng of DNA, 1 × ddPCR Supermix for Probes (BioRad, Hercules, CA, USA), 0.25 µM of each primer, and 0.125 µM of the probe in a total volume of 22  µL followed by droplet generation using an automated droplet generator (BioRad). The sequences of the EBV primer and probe set were as follows: forward primer, 5′-GCAGCCGCCCAGTCTCT-3′; reverse primer, 5′-ACAGACAGTGCACAGGAGCCT-3′ and probe, 5′-FAM-AAAAGCTGGCGCCCTTGCCTG-TAM-3′. The PCR amplicon length is 83 bp [14]. The sequences of the TERT primer and probe set were as follows: forward primer, 5′-GGGTCCTCGCCTGTGTACAG-3′; reverse primer, 5′-CCTGGGAGCTCTGGGAATTT-3′ and probe, 5′-VIC-CACACCTTTGGTCACTC-MGB-3′. The PCR amplicon length is 60 bp [13]. Cycling conditions included preheating at 95 °C for 10 min followed by 40 cycles of denaturation at 94 °C for 30 s, annealing at 56 °C for 60 s, and final heating at 98 °C for 10 min. After amplification, the PCR plate was transferred to a QX100 droplet reader (BioRad), and fluorescence amplitude data were obtained by QuantaSoft software (BioRad).

### 2.3. EBER-1 in Situ Hybridization

The presence of EBV was determined using ISH with EBER-1, which is known to be present in large amounts in EBV-infected cells. EBER-1 was detected with a biotin-labeled 30-base oligomer, using previously described procedures [7]. Paraffin-embedded 4-mm sections were deparaffinized, rehydrated, predigested with pronase, prehybridized, and then hybridized overnight at 37 °C. After washing with 0.5 × saline sodium citrate, hybridization was detected using an avidin-biotin complex method according to the manufacturer’s instructions.

### 2.4. Statistical Analysis

The cut-off value of the ddPCR for diagnosing EBVaGC was determined by discriminant characteristic analysis, which was performed using StatFlex Ver.6 (Artec). The Mann-Whitney test was also used (Ekuseru-Toukei 2010 for Windows; Social Survey Research Information Co., Ltd., Tokyo, Japan), and each result was determined to be significantly different when *p* < 0.05.

## 3. Results

### 3.1. Setting the Cut-Off Value of ddPCR to Detect EBVaGC

We prepared 47 lesions of EBVaGC that showed an EBER1 signal in almost all gastric cancer cells by EBER1 ISH (Figure 1) and 47 EBV-negative controls. The median values of EBV-DNA load for the EBVaGC and EBV-negative control were 17.0 and 0.00308, respectively, for which the Mann-Whitney test indicated a statistically significant difference of *p* < 0.01 (Figure 2). A cut-off value of 0.032 was determined at which the sensitivity was 1. The median values were 35.6 and 6.40 for EBV-associated advanced cancer and early-stage cancer, respectively, and the EBV-DNA load was significantly higher in the EBV-associated advanced cancer than that in the early-stage cancer (Figure 2).

### 3.2. Evaluation of the Cut-Off Value of ddPCR

We prepared 158 samples from gastric cancers for which the existence of EBV was unknown. The EBER1 ISH method revealed an EBER1 signal in the nucleus of gastric cancer cells in 14 of the 158 lesions (8.9%). These 14 lesions were judged to be EBV-positive as the cut-off value of their EBV-DNA load was exceeded in each lesion. The results of the ddPCR and the EBER1 ISH were in complete agreement. The median values of EBV load for the EBVaGC and EBV-negative control samples were 4.75 and 0.0006, respectively, for which the Mann-Whitney test indicated a statistically significant difference of *p* < 0.01 (Figure 3). We showed typical results of ddPCR for EBVaGCs and EBV-negative gastric cancers (Figure 4a,b). Clinicopathological features of the EBV-positive and -negative cases are listed in Table 1.

### 3.3. Diagnosis of EBVaGC Using Endoscopic Biopsy Specimens

The median values of EBV load for EBVaGC and EBV-negative control were 9.55 and 0.0025, respectively, with the Mann-Whitney test indicating a statistically significant difference of *p* < 0.01. Even when using biopsy specimens as samples for ddPCR, the EBV-DNA load of all EBVaGCs was larger than the cut-off value, whereas the EBV-DNA load of the EBV-negative gastric cancers was small (Figure 5). 

### 3.4. Detection EBV-DNA from Serum Samples of EBVaGC

We also examined the detectability of EBV-DNA in blood samples from EBVaGC patients and found that EBV-DNA could be detected in the blood of patients with EBV-positive advanced gastric cancers. The median value of the EBV-DNA load was 0.018. However, it was difficult to evaluate the EBV-DNA load in EBV-positive early gastric cancers and EBV-negative gastric cancers as the median value in those patients was zero (Figure 6).

## 4. Discussion

The EBER1 ISH method, which has excellent sensitivity and specificity, has been used to diagnose EBVaGC. EBER1 is a 170b small non-coding RNA that is present in EBV-infected cells and can be stained even in formalin-fixed paraffin-embedded samples [7,8]. However, the detection method for EBV infection is expensive and time consuming and has not been applied to preoperative diagnosis. The EBV is a DNA virus, and the BamH1-digested W fragment has been targeted for PCR because it has a repeat sequence [14]. There have been reports of applying PCR to the diagnosis of EBVaGC [15]. The detection sensitivity of EBV-DNA by PCR was 100%, but the specificity was low. It is thought that EBV infection in non-cancerous mucosa and lymphocytes could be detected as false positives [9]. In the present study, the diagnostic ability of ddPCR for EBV-DNA was completely consistent with the results of EBER1 ISH. The highly quantitative nature of ddPCR has shown that EBVaGC can be accurately diagnosed.

The clinical significance of EBVaGC has been reported. EBVaGC is classified into one of the molecular subtypes in TCGA, and overexpression of PD-L1 has been shown in this cancer [5]. The pathological characteristics of EBVaGC are shown by its infiltration of lymphocytes [16,17,18,19]. These findings were frequently observed in responders to immune checkpoint inhibitors [20,21]. Thus, the efficacy of immune checkpoint inhibitors against EBVaGC is expected [6]. In a prospective phase II clinical trial of pembrolizumab, dramatic responses were actually observed in patients with EBVaGC and MSI-H tumors [22]. Unresectable, recurrent gastric cancer that can be used for chemotherapy such as anti-PD-1 antibody often provides only biopsy tissue, therefore, our ddPCR to diagnose EBVaGC with DNA from biopsy tissue is useful. We believe that our diagnostic method using ddPCR could be a companion test for EBVaGC to select an appropriate chemotherapy.

Using the EBV-DNA load in blood as a biomarker for EBV-related diseases has already been attempted. Chan et al. used the detection of EBV-DNA in plasma to screen for nasopharyngeal carcinoma in 20,174 Chinese patients [23]. They found 309 patients (1.5%) with detectable EBV-DNA in plasma, and 34 of these patients (0.17% of all patients) had nasopharyngeal carcinoma on endoscopic evaluation, whereas only 1 of the EBV-DNA-negative patients presented with nasopharyngeal carcinoma. Overall, the sensitivity and specificity of this approach were 97.1% and 98.6%, respectively. In EBVaGC, the detection sensitivity and specificity of real-time PCR analysis were 71.4% (10/14) and 97.1% (135/139), respectively [24]. Qiu et al. showed that the positive rate of plasma EBV-DNA in EBVaGC was 43.6% (61/140), and the plasma EBV-DNA loads significantly increased with the advancement in TNM stages [25]. By using ddPCR, we could detect EBV-DNA in blood in advanced EBVaGC. These results suggest that assessment of the EBV-DNA load in blood can diagnose advanced EBVaGC and predict the courses of treatment of EBVaGC. 

## 5. Conclusions

We established a new diagnostic method for EBVaGC from tissue samples by ddPCR. Measurement of the EBV-DNA load in blood by ddPCR might be able to diagnose EBVaGC and evaluate the treatment courses of EBVaGC.

## Figures and Tables

**Figure 1 microorganisms-07-00628-f001:**
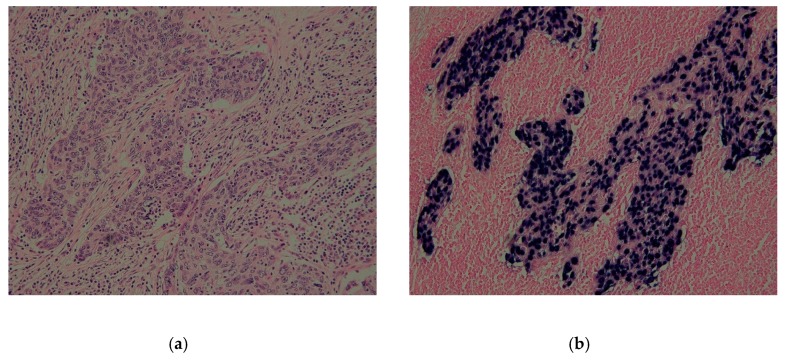
Histologic findings of EBVaGC. (**a**) H&E staining showed massive infiltration of lymphocytes with undifferentiated adenocarcinoma (× 200). (**b**) EBER1 in situ hybridization showed that signals of EBER1 were observed in almost all of the cancer cells (× 200). *EBVaGC* Epstein-Barr virus-associated gastric carcinoma, *H&E* hematoxylin and eosin stain.

**Figure 2 microorganisms-07-00628-f002:**
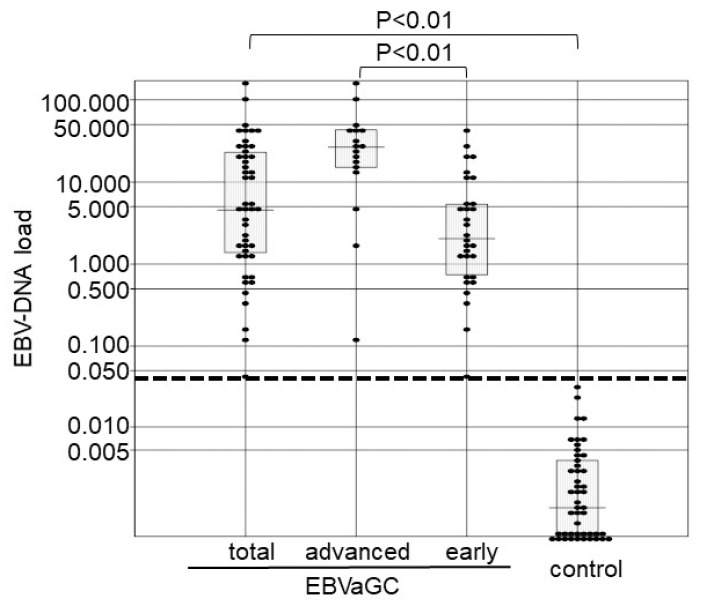
EBV-DNA load in the control, early stage, advanced stage, and total EBVaGC groups. Each sample is indicated by a closed circle. Box plots show the median with interquartile range (25th and 75th percentile), and the dotted line represents a cut-off value of 0.032. *EBVaGC* Epstein-Barr virus-associated gastric carcinoma.

**Figure 3 microorganisms-07-00628-f003:**
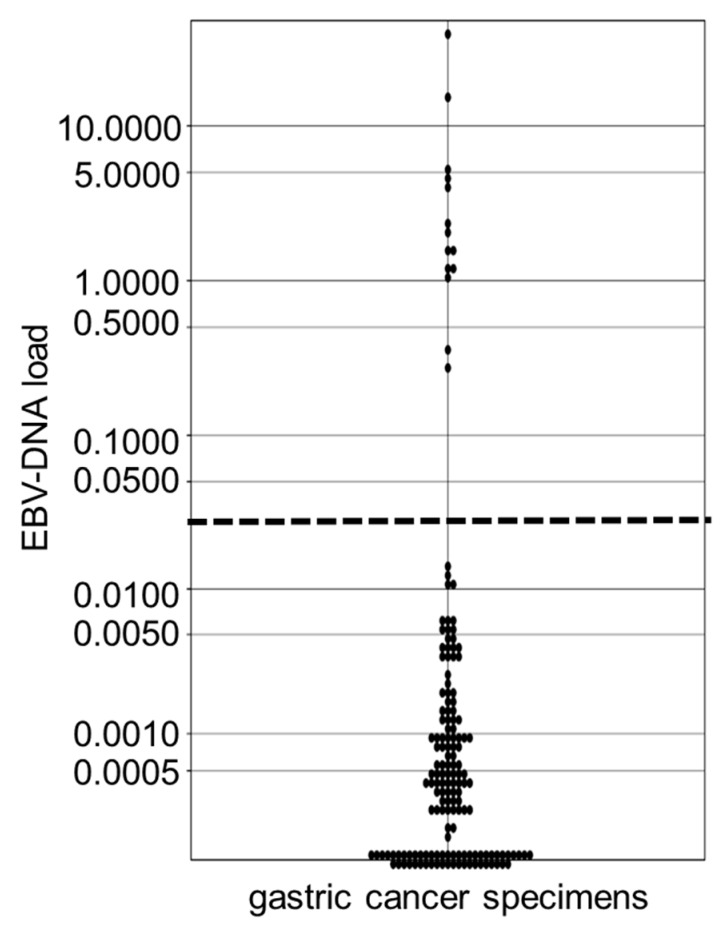
EBV-DNA load in the gastric cancer groups whose EBV status is unknown. Each sample is indicated by a closed circle. The dotted line represents a cut-off value of 0.032. *EBV* Epstein-Barr virus.

**Figure 4 microorganisms-07-00628-f004:**
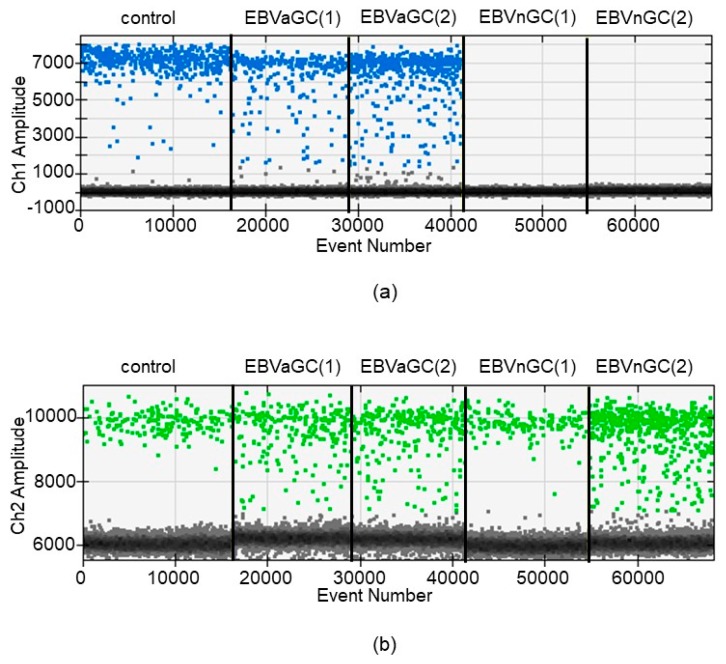
Representative results of EBV-DNA load by ddPCR. (**a**) The ddPCR results for detection of BamH1-W fragment of EBV in positive control, two EBVaGCs and two EBV-negative GCs. Positive droplets with PCR amplification are shown in blue; negative droplets without any amplification are shown in grey. Five ddPCR reactions are divided by the vertical black lines. (**b**) The ddPCR results for detection of TERT in positive control, two EBVaGCs and two EBV-negative GCs. Positive droplets with PCR amplification are shown in green and negative droplets without any amplification are shown in grey. Five ddPCR reactions are divided by the vertical black lines. *EBVaGC* Epstein-Barr virus-associated gastric carcinoma, *ddPCR* droplet digital polymerase chain reaction, *TERT* telomerase reverse transcriptase.

**Figure 5 microorganisms-07-00628-f005:**
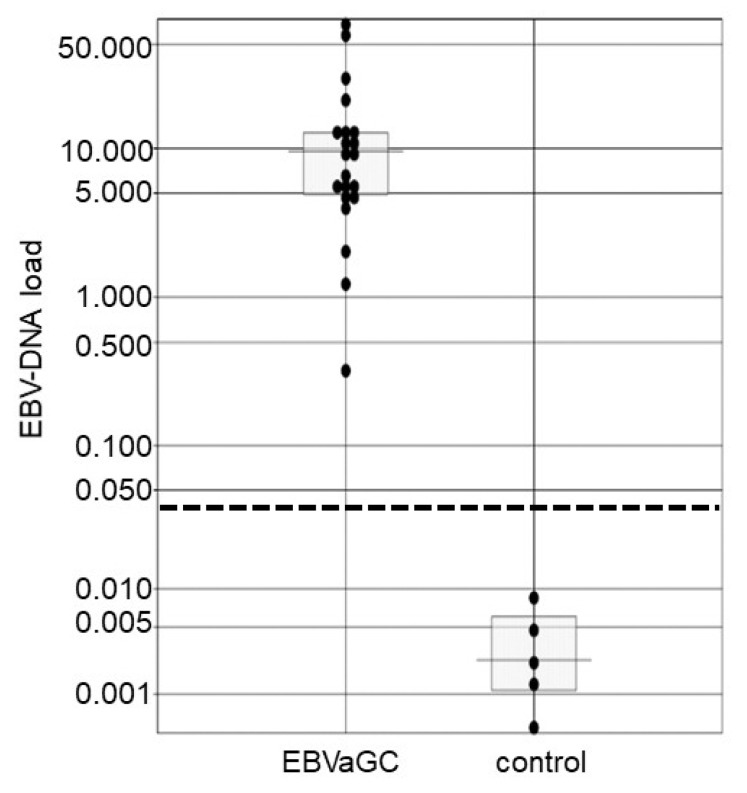
EBV-DNA load in specimens from endoscopic biopsies of the control and EBVaGC groups. Each sample is indicated by a closed circle. Box plots show the median with interquartile range (25th and 75th percentile), and the dotted line represents a cut-off value of 0.032. *EBVaGC* Epstein-Barr virus-associated gastric carcinoma.

**Figure 6 microorganisms-07-00628-f006:**
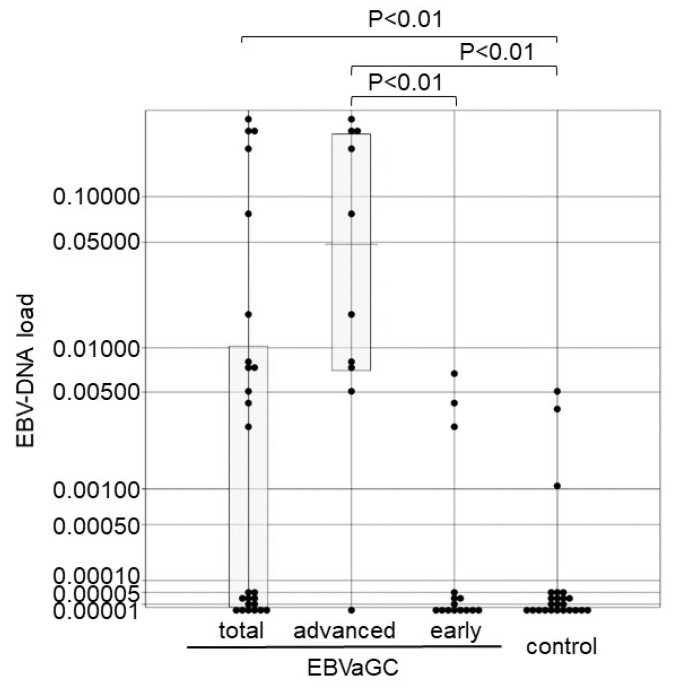
EBV-DNA load in the serum samples of the control and EBVaGC groups. Each sample is indicated by a closed circle. Box plots show the median with interquartile range (25th and 75th percentile). *EBVaGC* Epstein-Barr virus-associated gastric carcinoma.

**Table 1 microorganisms-07-00628-t001:** Cases for validation of the cut-off value of EBV-DNA load.

Total	*n* = 158	EBV-positive(*n* = 14)	EBV-negative (*n* = 144)
Age, years, mean (range)	67.1(28–92)	64.9(53-76)	67.1(35-92)
Sex			
Male	101	13	88
Female	57	1	56
Histologic type			
Differentiated type	68	6	62
Undifferentiated type	90	8	82
Depth of tumor invasion			
Early stage	76	9	67
Advanced stage	82	5	77
*EBV* Epstein-Barr virus

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
