# Peer review of "Establishment of a Screening Method for Epstein-Barr Virus-Associated Gastric Carcinoma by Droplet Digital PCR"

_microorganisms, 2019, doi:10.3390/microorganisms7120628_

Round 1

Reviewer 1 Report

The paper entitled "Establishment of a screening method for Epstein-Barr virus-associated gastric carcinoma by droplet digital PCR" is an interesting study of using droplet digital PCR (ddPCR) as a novel method to enable the highly sensitive and quantitative detection of EBV. An in situ hybridization (ISH) method for EBV-encoded small RNA1 (EBER1) has commonly been used to detect EBV. EBVaGC is defined as a gastric carcinoma showing EBER1 signals in the nuclei of almost all carcinoma cells detected by ISH, however EBER-ISH is expensive and time consuming and have not been applied to preoperative diagnosis. Real-time quantitative PCR was applied to examine the association between the copy number of EBV-DNA and the clinical courses of EBV-associated diseases. The authors conducted both ddPCR and EBER1 ISH to examine whether their results coincided in 158 gastric cancer specimens of unknown EBV status. They prepared 26 biopsy specimens and 49 serum samples including EBVaGC and assayed them by ddPCR. Obtained results were as follows: 1/ the mean values of EBV-DNA load for EBVaGC and EBV-negative control were 17.0 and 0.00308, respectively; a cut-off value of 0.032 was determined for which the sensitivity was 1; 2/among the 158 gastric cancer specimens, 14 lesions were judged as EBV-positive by the 0.032 cut-off value determined by ddPCR; 3/ the results of ddPCR and EBER1 ISH were in complete agreement. Even when using a biopsy specimen as a sample for ddPCR, the EBV-DNA load of all EBVaGCs was larger than the cut-off value. The authors established a new method of diagnosing EBVaGC from tissue samples by ddPCR, which is cost-effective and may be useful in cancer diagnosis and treatment. Therefore, I recommend to publish this paper in its present form.

Author Response

Thank your for reviewing our manuscript.

Reviewer 2 Report

Overall, this is a very interesting study that has the potential to effect a step-change in the diagnosis of EBV-positive cancers by offering a cheaper and quicker alternative to in situ hybridization, the current gold standard in the field.

There are a handful of typographical errors throughout – please refer to the list below to make amendments.

In terms of the results section, I have a few queries:

Figure 1 – why have the authors opted for black and white images for this figure? I personally feel a colour photo would be more preferable and would make the interpretation, especially of figure 1a, easier for the reader. Table 1 – there appear to be no values for “Depth of tumor invasion”? Or is this a subtitle within the table? Please clarify. Also, “EBV Epstein-Barr virus” at the bottom of the table seems a little out of place. Is this meant to be a descriptive term as you would see in a legend or description? If so, it does not belong as part of the table itself. Please rectify. Subsection 2.3, line 158 – the authors state, “EBVaGC and EBV-negative control were 9.55 and 0.0025, respectively…” – 9.55 and 0.0025 what? Please provide clarity around this statement so that it’s more easily understandable for the reader (I’m assuming you’re referring to copy number, but this needs to be made more clear)

The discussion section feels a little limited in places. For example, in the second paragraph (lines 188-198) the authors begin to discuss the implications of overexpression of PD-L1 in EBVaGC, but doesn’t really draw the point together with their current findings and draw conclusions about its significance. Again, I’m left to make the assumption that by identifying EBV-positive GC using ddPCR one can more easily identify suitable candidates for PD-L1 inhibitor immunotherapy, but this is not made overtly clear. The same goes for the subsequent sentences regarding MSI.

On line 208 the authors claim “This result is consistent with our findings.” However, I’m not convinced this is so. How are they consistent? Are you referring to copy number? % positivity? Or EBV-DNA load with advanced stage disease? Please add a clarifying statement to this sentence.

On line 211 the authors mention the possibility of combining the detection of EBV-DNA from biopsy tissue and blood samples, but don’t expand on the possible benefits of doing so. Please consider adding to this to improve the overall quality of the discussion.

The conclusion is also rather brief – consider fleshing this out – for example, to tie in your diagnostic method with the implications for therapy.

List of typographical/grammatical errors:

Line 56 – too many uses of the word “and” in this sentence. Consider adding a comma after “(Janus activating kinase 2),” and removing the “and” between “…kinase 2),” and “PD-L1” Line 64 – please replace “have” with “has” to be grammatically correct Line 110 – please define “SSC” on first use Line 168 – this sentence lacks clarity. Please consider rephrasing to “…could be detected in the blood of EBV-positive advanced gastric cancer patients.” Line 178 – please add “non-coding” between “small” and “RNA” (so it reads “small non-coding RNAs” Line 180 – please replace “have” with “has” to be grammatically correct Line 183-184 – please consider rephrasing to “It is thought that EBV infection in non-cancerous mucosa and lymphocytes could be detected as false positives.” Line 186 – please replace “high” with “highly” to be grammatically correct Line 197 – please replace “an” with “a” to be grammatically correct Line 198 – please replace “adequate” with “appropriate” for clarity Line 204 – please replace “were” with “was” to be grammatically correct

Author Response

Thank you for reiviewing our manuscript and giving us valuable comments.

For Figure 1, we used colour images. We corrected Table 1 as shown in revised version of the manuscript. We corrected the sentence in Subsection 2.3, line 158 as your recommendation.  In the discussion, we added explanation that our ddPCR could be useful for selection for appropriate chemotherapy of EBVaGC. "Unresectable, recurrent gastric cancer that can be used for chemotherapy such as anti-PD-1 antibody often provides only biopsy tissue therefore our ddPCR to diagnose EBVaGC with DNA from biopsy tissue is useful. We believe that our diagnostic method using ddPCR could be a companion test for EBVaGC to select an appropriate chemotherapy." We deleted the sentence “This result is consistent with our findings.” We mentioned that assessment of the EBV-DNA load in blood can know the clinical courses of EBVaGC in the discussion section and conclusion. We corrected some errors as your suggestion.